# NDRG1 activates VEGF-A-induced angiogenesis through PLCγ1/ERK signaling in mouse vascular endothelial cells

Kosuke Watari[1], Tomohiro Shibata[1], Hideaki Fujita[2], Ai Shinoda[1], Yuichi Murakami[1,3], Hideyuki Abe[4], Akihiko Kawahara[4], Hiroshi Ito[1,5], Jun Akiba[4], Shigeo Yoshida[6], Michihiko Kuwano[3] & Mayumi Ono[1✉]

Many diseases, including cancer, have been associated with impaired regulation of angiogenesis, of which vascular endothelial growth factor (VEGF)-A is a key regulator. Here, we test the contribution of N-myc downstream regulated gene 1 (NDRG1) to VEGF-A-induced angiogenesis in vascular endothelial cells (ECs). $Ndrg1^{-/-}$ mice exhibit impaired VEGF-A-induced angiogenesis in corneas. Tumor angiogenesis induced by cancer cells that express high levels of VEGF-A was also reduced in a mouse dorsal air sac assay. Furthermore, NDRG1 deficiency in ECs prevented angiogenic sprouting from the aorta and the activation of phospholipase Cγ1 (PLCγ1) and ERK1/2 by VEGF-A without affecting the expression and function of VEGFR2. Finally, we show that NDRG1 formed a complex with PLCγ1 through its phosphorylation sites, and the inhibition of PLCγ1 dramatically suppressed VEGF-A-induced angiogenesis in the mouse cornea, suggesting an essential role of NDRG1 in VEGF-A-induced angiogenesis through PLCγ1 signaling.

[1] Department of Pharmaceutical Oncology, Graduate School of Pharmaceutical Sciences, Kyushu University, Fukuoka 812-8582, Japan. [2] Faculty of Pharmaceutical Sciences, Nagasaki International University, Sasebo 859-3243, Japan. [3] Cancer Translational Research Center, St. Mary's Institute of Health Sciences, Kurume 830-8543, Japan. [4] Department of Diagnostic Pathology, Kurume University Hospital, Kurume 830-0011, Japan. [5] Department of Neurosurgery, Faculty of Medicine, Saga University, Saga 849-8501, Japan. [6] Department of Ophthalmology, Kurume University School of Medicine, Kurume 830-0011, Japan. ✉email: mono@phar.kyushu-u.ac.jp

Angiogenesis is essential for normal development, tissue homeostasis, and repair, and is tightly regulated by angiogenic factor-induced activation of angiogenic signaling. The disruption of this regulation is followed by abnormal angiogenesis, contributing to a wide variety of diseases, including diabetic retinopathy, age-related macular degeneration, and cancer[1]. VEGF-A, a representative key regulator of angiogenesis under physiological and pathological conditions[2], binds to VEGF receptor 2 (VEGFR2) during the early process of VEGF-A signaling, resulting in its phosphorylation and internalization via endocytic vesicles, followed by the activation of PLCγ1[3,4]. Furthermore, during the late process of VEGF-A signaling, PLCγ1, after migrating to the cell surface membrane, enhances the expression levels of inositol 1,4,5-trisphosphate (IP$_3$), and diacylglycerol (DAG), and switches on the protein kinase C (PKC)/extracellular signal-regulated protein kinases 1 and 2 (ERK1/2) signaling pathway[5,6]. Of these VEGF-A signaling roles, the mechanism underlying how PLCγ1 turns on the downstream signaling is not fully understood.

The N-myc downstream regulated gene 1 (NDRG1) is a member of the evolutionarily well conserved NDRG gene family containing an α/β hydrolase fold and a C-terminal domain harboring glycogen synthase kinase (GSK)-3β and serum and glucocorticoid-regulated kinase (SGK) phosphorylation sites[7–10]. NDRG1 is ubiquitously expressed and, in each cell or tissue type, plays several important roles in growth, development, and differentiation[11–13]. In various cell and tissue types, it is involved in multiple stages of differentiation, including placentation[14], trophoblast formation[15], mast cells[16], neutrophils[17], and macrophage lineage cell maturation[18].

On the other hand, previous studies have demonstrated the effect of enhanced expression of NDRG1 on tumor angiogenesis and on its negative impact on the outcomes of cancer patients, in various tumors, including prostate cancer[19], pancreatic cancer[20,21], cervical cancer[22], lung cancer[23], and stomach cancer[24,25]. We have further reported that an NDRG1 deficiency impairs the production of VEGF-A and other angiogenic factors from macrophages, thereby attenuating tumor angiogenesis and inflammatory-induced angiogenesis[18]. In addition, NDRG1 was originally identified as tunicamycin-responsive protein, which is enhanced by homocysteine treatment of cultured human vascular endothelial cells (ECs)[26]. However, whether NDRG1 in ECs could play any role in angiogenic response to angiogenic factors, including VEGF-A, remains unclear.

In this study, aiming to understand the essential role of NDRG1 in ECs in response to the potent angiogenic factor VEGF-A, we assessed the effect on angiogenesis of NDRG1 deficiency in ECs in response to VEGF-A and another potent angiogenic factor fibroblast growth factor (FGF)-2. Furthermore, we observed that in ECs of human tumor tissues, NDRG1 expression was highly rich. Here, to our knowledge, we present the specific and novel role of NDRG1 in VEGF-A-induced angiogenesis in close association with PLCγ1 activation.

## Results

**VEGF-A-induced angiogenesis is impaired in $Ndrg1^{-/-}$ mice.** To determine which cells highly express NDRG1 in angiogenesis-related disease such as cancer, we first investigated the expression of NDRG1 of human tumor tissues using immunofluorescent analysis. Figure 1a shows co-expression of both vascular EC marker CD34 and NDRG1 in tumor tissues from five patients with breast cancer. Interestingly, NDRG1 is shown to be rather more abundantly expressed in ECs than in other cell types including cancer cells.

To elucidate whether and how NDRG1 plays some essential roles in angiogenesis, we next examined whether NDRG1 plays any angiogenic role by ablation of $Ndrg1$ gene ($Ndrg1^{+/+}$ vs $Ndrg1^{-/-}$ mice). We previously reported that tumor growth was significantly retarded accompanied with impaired angiogenesis in $Ndrg1^{-/-}$ mice[18]. In subsequent experiments, we performed dorsal air sac assays in $Ndrg1^{+/+}$ and $Ndrg1^{-/-}$ mice to investigate the angiogenic activities in response to cancer cell-derived angiogenic factors. Using quantitative real-time polymerase chain reaction (PCR), we determined the expression levels of VEGF-A in mouse melanoma B16/BL6, mouse lung cancer LLC/3LL, and murine renal cell carcinoma (RENCA) cells. Expression of VEGF-A was about eightfold higher in RENCA cells than that in B16/BL6 and LLC/3LL cells (Fig. 1b). After implanting chambers containing RENCA cells that produce abundant VEGF-A, we compared tumor angiogenesis between $Ndrg1^{+/+}$ and $Ndrg1^{-/-}$ mice using dorsal air sac assay. Tumor angiogenesis, as indicated by irregular formations of new blood vessels, was strongly induced in $Ndrg1^{+/+}$ mice. In contrast, tumor angiogenesis was only slight in $Ndrg1^{-/-}$ mice under the same conditions (Fig. 1c).

In further investigations into the angiogenic roles of NDRG1, we monitored angiogenic responses to the potent angiogenic factors VEGF-A and FGF-2 in corneas of $Ndrg1^{+/+}$ and $Ndrg1^{-/-}$ mice. In these experiments, VEGF-A-induced neovascularization in mouse corneas was almost abolished in $Ndrg1^{-/-}$ mice compared with that in $Ndrg1^{+/+}$ mice (Fig. 1d). However, following stimulation with FGF-2, we observed similar levels of neovascularization in $Ndrg1^{+/+}$ and $Ndrg1^{-/-}$ mice (Fig. 1e). Together, these data suggested that NDRG1 in the host plays an essential role in VEGF-A-induced angiogenesis.

**$Ndrg1$ deficiency in ECs impairs VEGF-A-induced angiogenesis.** Next, we examined the contribution of NDRG1 in ECs to VEGF-A-induced angiogenesis by using aortic ring assay (Fig. 2a, b). Following stimulation with VEGF-A, a far lower level of vascularization was induced in aortic rings from $Ndrg1^{-/-}$ mice than that induced in $Ndrg1^{+/+}$ mice, whereas FGF-2-induced angiogenesis was similarly observed in aortic rings of $Ndrg1^{+/+}$ and $Ndrg1^{-/-}$ mice (Fig. 2b).

We further examined whether cell proliferation and migration in mouse ECs (mECs) isolated from $Ndrg1^{+/+}$ and $Ndrg1^{-/-}$ murine lung tissues are impaired by $Ndrg1$ deficiency following treatments with VEGF-A or FGF-2. The purity of mECs were verified the expression of both CD31 and ICAM-2 by the flow cytometric analysis (CD31-positive cells: P1 = 95.65 ± 0.5248%, P3 = 88.4475 ± 1.0785%; ICAM-2-positive cells: P1 = 74.315 ± 9.3011%, P3 = 70.905 ± 1.2619%), and passage number of mECs in all experiments was equal to or less than three times (Supplementary Fig. 1a, b). We observed no morphological differences between mECs from $Ndrg1^{+/+}$ (WT-mECs) and $Ndrg1^{-/-}$ mice (KO-mECs) under basal growth conditions (Supplementary Fig. 2a). WT-mECs and KO-mECs had similar proliferation rates, with doubling times of about 20 h under basal growth conditions (Supplementary Fig. 2b). However, following stimulation with VEGF-A, we observed significant (**$P < 0.01$) increases in cell proliferation only in WT-mECs (Fig. 2c). Consistent with Figs. 1e and 2b, FGF-2 significantly (*$P < 0.05$, **$P < 0.01$) stimulated cell proliferation in mECs from both $Ndrg1^{+/+}$ and $Ndrg1^{-/-}$ mice (Fig. 2c). Furthermore, VEGF-A stimulated cell migration only in WT-mECs, whereas FGF-2 stimulated similar cell migration activities in both WT- and KO-mECs (Fig. 2d). These data indicate the lack of angiogenic responses to VEGF-A in KO-mECs, whereas angiogenic responses to FGF-2 remain intact.

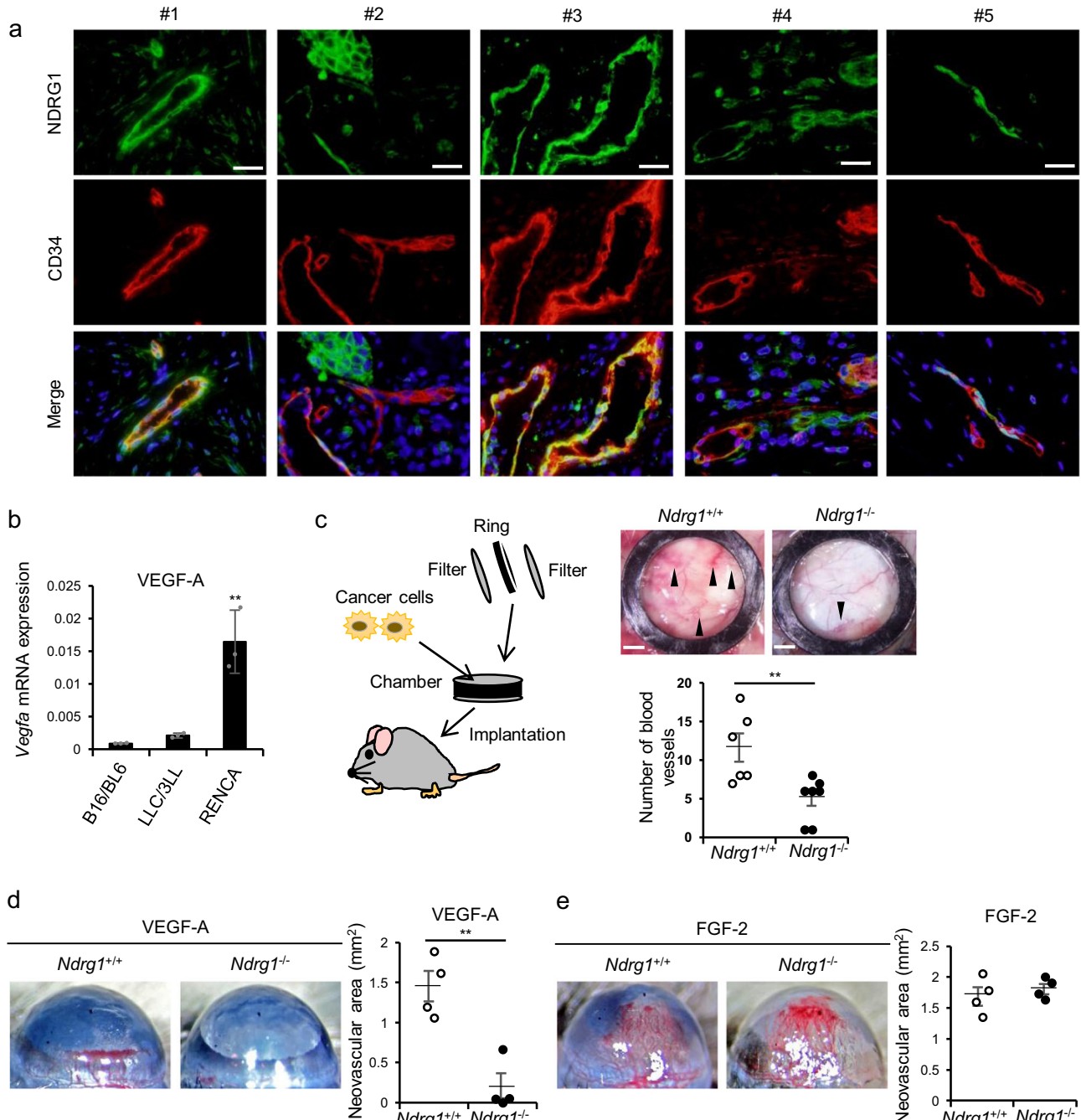

**Fig. 1 Ndrg1 deficiency impairs VEGF-A-induced angiogenesis in mice. a** Immunofluorescent images of NDRG1 (green) and CD34 (red) using a specific antibody in five surgically resected breast cancer specimens. ×400 original magnification; scale bar = 10 μm. **b** We determined *Vegfa* mRNA expression in mouse cancer B16/BL6, LLC/3LL, and RENCA cells by qRT-PCR (*n* = 3 per group). We normalized the data by *Gapdh* expression levels. **c** Left: Schematic illustration of experimental mouse dorsal air sac assay. Right: We evaluated tumor angiogenesis by RENCA cells using mouse dorsal air sac assays. We determined angiogenic responses by counting numbers of newly formed blood vessels of >3 mm in length (*Ndrg1*[+/+] group, *n* = 6; *Ndrg1*[−/−] group, *n* = 7). Arrowheads indicate newly formed vessels with characteristic zigzagging lines; scale bar = 2 mm. **d, e** We induced corneal neovascularization by VEGF-A (200 ng/pellet; **d**) or FGF-2 (100 ng/pellet; **e**) on day 7 after implanting Hydron pellets into mouse corneas (original magnification ×20). We performed quantitative analysis of neovascularization of each group on day 7. Areas are expressed in mm² (*n* = 4 mice per group). We present data as means ± SE (**c–e**), and SD (**b**) of *n* observations, and we identified differences using two-tailed *t*-test, \*\**P* < 0.01.

**NDRG1 deficiency suppresses VEGF-A signaling pathway in ECs**. To investigate the mechanisms underlying the effects of NDRG1 in ECs on VEGF-A-induced angiogenesis, we determined the effects of *Ndrg1* deficiency on the downstream signaling molecules, AKT and ERK1/2, both closely associated with VEGF-A-induced angiogenesis[27,28]. Treatments with exogenous

VEGF-A stimulated phosphorylation of ERK1/2 in WT-mECs but not in KO-mECs (Fig. 3a). However, we observed increases in ERK1/2 phosphorylation following treatments of WT- and KO-mECs with FGF-2 (Fig. 3b). Furthermore, the time kinetics of VEGF-A-induced ERK1/2 phosphorylation were significantly (\**P* < 0.05) stronger in WT-mECs than in KO-mECs, whereas in

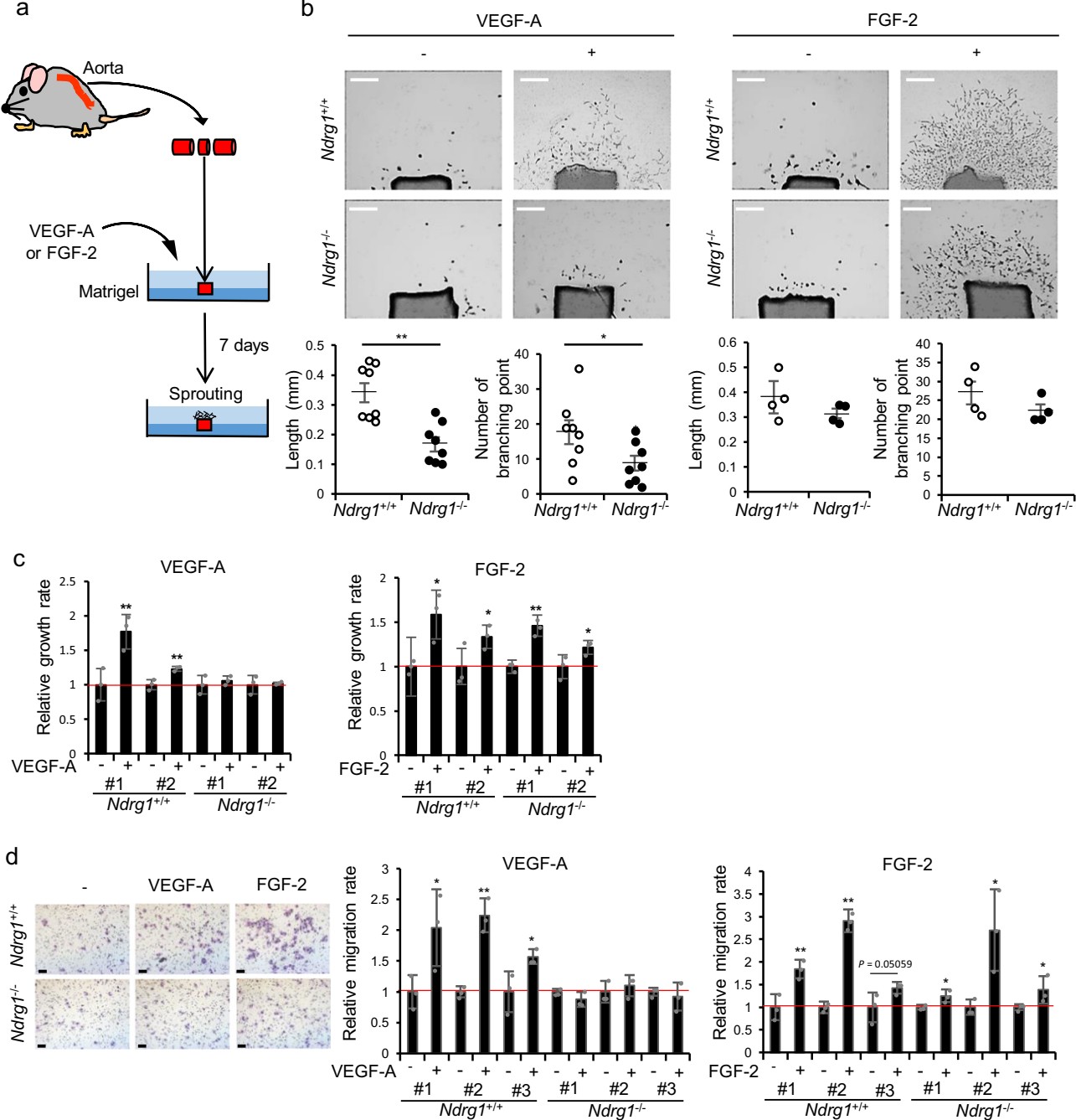

**Fig. 2 Ndrg1 deficiency in endothelial cells selectively impairs VEGF-A-induced angiogenesis. a** Schematic illustration of experimental mouse aortic ring assay. **b** Photographs of aortic rings after incubation for 7 days with or without 25 ng/ml VEGF-A or 50 ng/ml FGF-2. Lengths (left graph) and numbers of branching points (right graph) of aortic ring sprouts are indicated: $n = 8$ per group (VEGF-A); $n = 4$ per group (FGF-2); scale bar = 0.5 mm. **c** Relative cell growth rates of ECs from $Ndrg1^{+/+}$ and $Ndrg1^{-/-}$ mice in the presence or absence of 20 ng/ml VEGF-A (left) or 10 ng/ml FGF-2 (right) over 48 h. We counted cells using a Coulter counter and normalized numbers of cells in the presence of angiogenic factors to those in untreated cultures (1.0); we isolated #1 and #2 of each group from independent mice ($n = 3$). **d** We determined relative cell migration rates of ECs from mice in the presence and absence of treatments with 50 ng/ml VEGF-A (left) or 20 ng/ml FGF-2 (right) for 6 h; scale bar = 0.1 mm. We manually counted cells that migrated under the filter using a microscope and normalized numbers of cells in treatment groups to those in control cultures (1.0). We isolated #1, #2, and #3 of each group from independent mice ($n = 3$). We present data as means ± SE (**b**) and SD (**c**, **d**) of $n$ observations, and identified differences using one- (**c**, **d**) or two-tailed (**b**) $t$-test, $*P < 0.05$, $**P < 0.01$.

both mECs VEGFR2 phosphorylation by VEGF-A was stimulated to similar levels (Fig. 3c). In WT-mECs, but not in KO-mECs, AKT phosphorylation was also stimulated by VEGF-A (Fig. 3a, c). By contrast, FGF-2-stimulated AKT phosphorylation levels in both WT- and KO-mECs were similar (Fig. 3b).

To further investigate the effects of *NDRG1* silencing on angiogenic signaling, we performed experiments in human ECs (hECs) with cognate siRNA. VEGF-A stimulated VEGFR2 phosphorylation to similar levels in siControl- and siNDRG1-treated hECs (Fig. 3d). However, at 5 min, VEGF-A stimulated

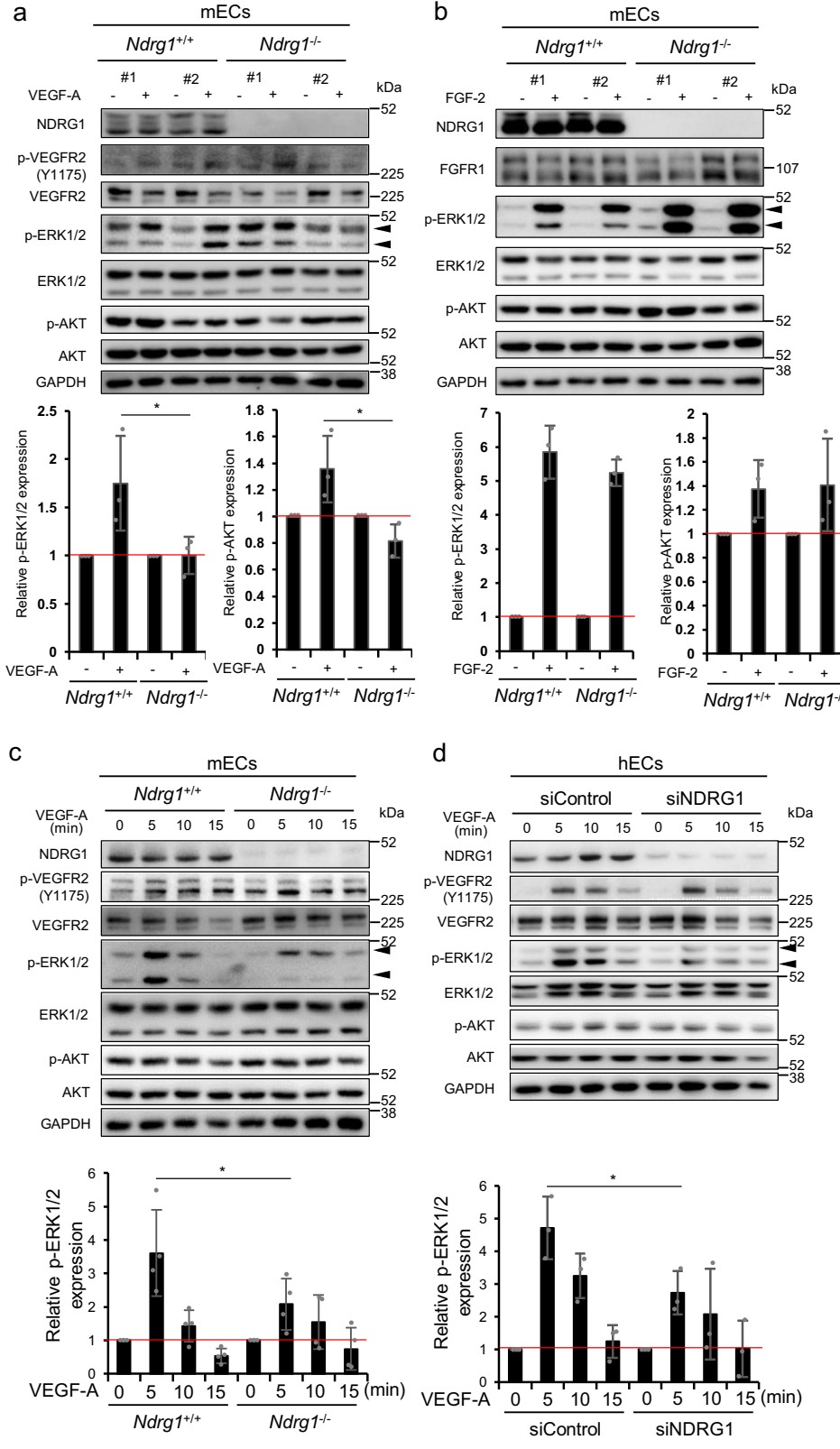

ERK1/2 phosphorylation to a significantly ($*P < 0.05$) reduced level in siNDRG1-treated hECs than in siControl-treated hECs (Fig. 3d). VEGF-A also augmented AKT phosphorylation in siControl-treated hECs in a time-dependent manner, but not in siNDRG1-treated hECs (Fig. 3d).

**NDRG1 does not affect the expression and function of VEGFR2.** VEGF-A, during multiple steps of its signaling, initially interacts with its cognate receptor VEGFR2 on the cell surface of ECs, and follows the subsequent endocytosis and cycle-back to the cell surface of the complex, during the early part of the

**Fig. 3 *NDRG1* deficiency impairs VEGF-A-driven ERK1/2 activation in ECs. a**, **b** Western blot analyses show protein expression levels of the angiogenesis receptor, ERK1/2 and AKT in mECs from lung tissues after culture in serum-free medium for 24 h and stimulation with 50 ng/ml VEGF-A (**a**) or 10 ng/ml FGF-2 (**b**) for 7 min. We established #1 and #2 for each group from independent mice. ERK1/2 and AKT phosphorylation levels in the presence of VEGF-A or FGF-2 were normalized to those in unstimulated cells (1.0) ($n = 3$ per group). **c** Kinetics of VEGFR2, ERK1/2, and AKT phosphorylation in mouse lung ECs after addition of 50 ng/ml VEGF-A. Western blot analyses show phosphorylation of VEGFR2, ERK1/2, and AKT in mECs from lung tissues after culture in serum-free medium for 24 h and stimulation with 50 ng/ml VEGF-A at indicated times ($n = 4$). We normalized phosphorylated ERK1/2 in the presence of VEGF-A to that at 0 min (1.0). **d** Western blot analyses show phosphorylation of VEGFR2, ERK1/2, and AKT in HUVECs following siRNA-mediated *NDRG1* silencing, and we quantified ERK1/2 phosphorylation following stimulation with VEGF-A ($n = 3$). We normalized phosphorylated ERK1/2 in the presence of VEGF-A to that at 0 min (1.0). We present data as means ± SD of $n$ observations, and identified differences using one- (**c**) or two-tailed (**a**, **b**, **d**) $t$-test, *$P < 0.05$. **a–d** Source data are provided as Supplementary Fig. 3.

process[3] (Fig. 4a). We examined whether *NDRG1* deficiency interferes with this early process in ECs. In both WT-mECs and KO-mECs, the flow cytometric analysis showed similar expression levels of membrane-bound VEGFR2 (Fig. 4b), and the binding of VEGF-A to VEGFR2 was unaffected by *Ndrg1* deficiency (Fig. 4c). Endocytosis of VEGF-A/VEGFR2 complexes, after stimulation of ECs with VEGF-A, control the specificity, amplitude, and duration of the downstream signaling events[4]. Therefore, we used biotinylation experiments to compare the endocytosis of VEGFR2 between WT-mECs and KO-mECs (Fig. 4d). These assays showed prominent internalized VEGFR2 in both WT-mECs and KO-mECs from 5 to 30 min after stimulation with VEGF-A, and we identified no differences between WT- and KO-mECs (Fig. 4d). In siControl- and siNDRG1-treated hECs, following stimulation with VEGF-A, VEGFR2 was also similarly internalized (Fig. 4e). After the endocytosis of the VEGF-A/VEGFR2 complex, VEGFR2 is cycled back to the cell surface of ECs via membrane trafficking of the endocytic vesicles[29] (Fig. 4a). In the presence of cycloheximide for 30 min, VEGF-A stimulation led to a profound reduction in total and surface VEGFR2 at similar levels in both siControl- and siNDRG1-treated hECs (Fig. 4f). Removal of VEGF-A allowed the recovery of surface VEGFR2 after 60 min in both hECs, and the expression levels of surface VEGFR2 were not affected by *NDRG1* silencing (Fig. 4f). Thus, *NDRG1* deficiency exerts only a limited effect, if any, on the early processes of VEGF-A signaling.

**NDRG1 deficiency interferes in PLCγ1 activation by VEGF-A.**
Next, we examined whether, in VEGF-A-stimulated ECs, *NDRG1* deficiency impairs the late VEGF-A signaling process after the intracellular endocytosis (Fig. 5a). In this process, VEGFR2 is first phosphorylated at Y1175 in response to VEGF-A, and subsequently phosphorylates PLCγ1, resulting in increased IP$_3$ and DAG levels[5]. Ca$^{2+}$ is released from the endoplasmic reticulum (ER) in response to IP$_3$ and, then, regulates the PKC–RAF–MAPK/ERK kinase (MEK)–ERK cascade[30,31]. To assess the involvement of NDRG1 in these late processes, we stimulated ECs with VEGF-A, and compared PLCγ1 phosphorylation. In mECs isolated from *Ndrg1*$^{-/-}$ mice (Fig. 5b), and also in *NDRG1* silencing hECs (Fig. 5c), these experiments showed significantly lower levels of VEGF-A-induced phosphorylation of PLCγ1 and ERK1/2 (Fig. 5b: *$P < 0.05$, Fig. 5c: **$P < 0.01$). Furthermore, VEGF-A, but not FGF-2, induced PLCγ1 phosphorylation (Fig. 5d), indicating that PLCγ1 activation is not required for FGF-2-induced ERK1/2 phosphorylation, as shown previously[32]. These data strongly indicate that *NDRG1* deficiency in ECs selectively interferes with PLCγ1 phosphorylation, when stimulated by VEGF-A, but not by FGF-2.

Because PLCγ1 activation increases cytoplasmic Ca$^{2+}$ levels[6], we compared cytoplasmic Ca$^{2+}$ levels in response to VEGF-A between siControl- and siNDRG1-treated hECs to corroborate the effect of *NDRG1* deficiency on PLCγ1 phosphorylation. VEGF-A markedly increased cytoplasmic Ca$^{2+}$ levels in

siControl-treated hECs (orange line) up to 7 min, but had a very much smaller effect in siNDRG1-treated hECs (gray line) (Fig. 5e). In contrast, FGF-2 did not increase cytoplasmic Ca$^{2+}$ levels in both siControl- (blue line) and siNDRG1-treated (yellow line) hECs (Fig. 5e). Moreover, irrespective of the activation of VEGFR2 and PLCγ1, the Ca$^{2+}$ chelator 1,2-bis(2-aminophenoxy) ethane-$N,N,N',N'$-tetraacetic acid tetrakis(acetoxymethyl ester) (BAPTA-AM) almost completely inhibited not only VEGF-A-induced Ca$^{2+}$ release from the ER (Fig. 5f) but also phosphorylation of PKCδ and ERK1/2 in hECs (Fig. 5g). Collectively, these data indicate the loss of VEGF-A-induced ERK1/2 activation due to *NDRG1* deficiency as attributable to the loss of intracellular Ca$^{2+}$ release. PKC, widely studied in the context of Ca$^{2+}$ release, is involved in multiple well-characterized signaling pathways, and we used the PKC activator phorbol 12-myristate 13-acetate (PMA). We further investigated the roles of NDRG1 in PKC-mediated ERK1/2 activation in ECs. In these experiments, phosphorylation of ERK1/2 and PKCδ were enhanced to similar levels by treatments of siControl- and siNDRG1-treated hECs with PMA without affecting PLCγ1 activation (Fig. 5h), indicating that *NDRG1* deficiency does not affect PKC–RAF–MEK–ERK signaling pathways. Collectively, these results suggest that NDRG1 regulates VEGF-A-induced angiogenesis through activation of PLCγ1 in ECs.

**Interaction of NDRG1 with PLCγ1 facilitates VEGF-A signaling.**
Next, we examined how NDRG1 activates late VEGF-A signaling in close context with PLCγ1 in ECs. As shown in Fig. 6a, co-immunoprecipitation (IP) assays revealed that NDRG1 forms a complex with PLCγ1 in hECs. The amino acid sequence of NDRG1 indicates the presence of a phosphopantetheine attachment, a prominent α/β hydrolase fold, and phosphorylation sites by GSK3β and SGK at its C-terminal domain[7–10] (Fig. 6b). In determining the NDRG1 domain for interaction with PLCγ1, we individually deleted the N-terminal domain (1–89AA) (glutathione $S$-transferase (GST)-Δ1), the α/β hydrolase fold domain (90–306AA) (GST-Δ2), and the total (307–394AA) or partial (326–350AA) phosphorylation sites on the C-terminal domain (GST-Δ3 and GST-Δ4) of NDRG1 (Fig. 6b), and performed GST-pulldown assay. Figure 6b showed that PLCγ1 bound to GST-NDRG1 and GST-Δ1 and Δ2 mutants, but not to GST-Δ3 and Δ4 mutants which were deleted total or partial phosphorylation sites at C-terminal domain, respectively (Fig. 6b). The data show that NDRG1 interacts with PLCγ1 through its phosphorylation sites in ECs. We further examined whether interaction of NDRG1 with PLCγ1 affected PLCγ1 phosphorylation in ECs. Transfection of GST-Δ4 mutant, which could not interact with PLCγ1, significantly (**$P < 0.01$) decreased VEGF-A-induced PLCγ1 phosphorylation compared with Mock or GST-NDRG1 transfection (Fig. 6c).

To further examine the localization of NDRG1 and PLCγ1 in response to VEGF-A, we separated membrane, cytosol and nuclear fractions and analyzed them by western blotting analysis

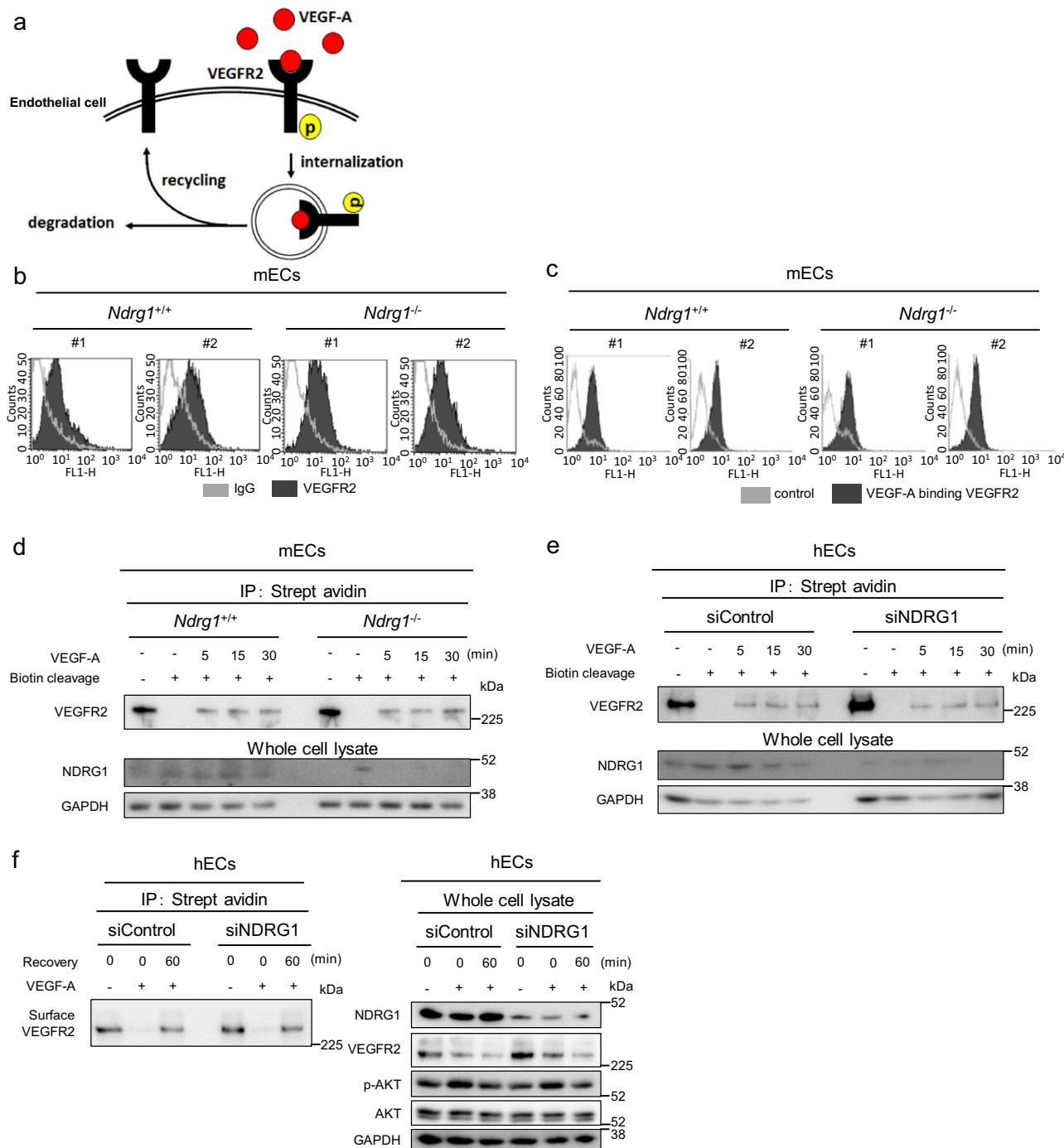

**Fig. 4 NDRG1 deficiency does not affect early process of VEGF-A signaling. a** Schematic image of early process of VEGF-A signaling, VEGF-A binds to VEGFR2 on the cell surface and, following endocytosis, is cycled back to the cell surface of the complex. **b, c** Flow cytometry analyses of cell surface (**b**) and VEGF-A-bound VEGFR2 expression (**c**) in mECs from *Ndrg1*$^{+/+}$ and *Ndrg1*$^{-/-}$ mice. We isolated #1 and #2 for each group from independent mice. **d** We labeled cell surface membranes of WT- and KO-mECs with cleavable biotin at 4 °C for 30 min and incubated them with VEGF-A (50 ng/ml) at 37 °C for indicated times, followed by cleavage of surface biotin. Then, we determined internalized biotinylated VEGFR2 using streptavidin bead pulldown and western blotting with an anti-VEGFR2 antibody. **e** We labeled cell surface membranes of HUVECs with cleavable biotin at 4 °C for 30 min and, then, incubated them with VEGF-A (20 ng/ml) at 37 °C for indicated times prior to cleavage of surface biotin. We determined the internalized biotinylated VEGFR2 using pulldown assays with streptavidin beads and western blotting with an anti-VEGFR2 assay. **f** We assessed VEGFR2 recycling in HUVECs after VEGF-A stimulation (50 ng/ml) for 30 min in the absence of protein synthesis (CHX inhibition) during the 60 min recovery phase after VEGF-A washout. **b–f** Source data are provided as Supplementary Figs. 3 and 4.

in hECs. Following treatment with VEGF-A, we observed the selective augmentation of PLCγ1 and NDRG1 in the membrane fraction in which activated PLCγ1 induced hydrolysis of phosphatidylinositol 4,5-bisphosphate to generate two second messengers, IP$_3$ and DAG (Fig. 6d). Then, we performed proximity ligation assay (PLA), which is an antibody-based method to allow in situ detection of endogenous protein interaction and localization with high specificity and sensitivity,

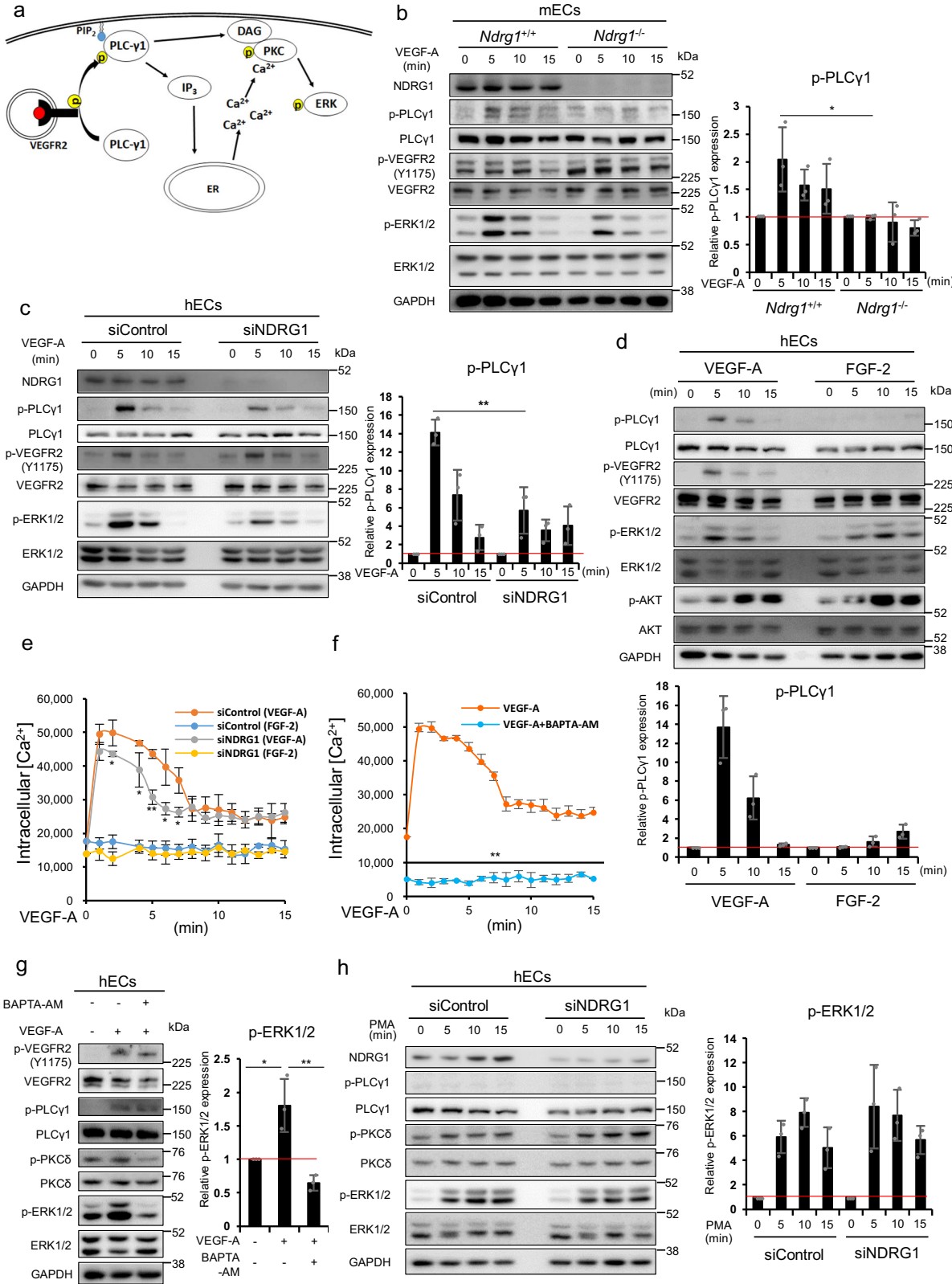

to further confirm the interaction and localization of NDRG1 and PLCγ1 in response to VEGF-A. Red fluorescent spots suggested the existence of a proximal localization and/or an interaction of NDRG1 and PLCγ1 in hECs in the absence and presence of VEGF-A (Fig. 6e). Furthermore, after VEGF-A stimulation for 5 min, PLA-positive spots significantly (**$P < 0.01$) increased around cell membrane (Fig. 6e). These data strongly suggested

that NDRG1 formed complex with PLCγ1, and followed by activation and translocation of PLCγ1 to cell membrane in response to VEGF-A (Fig. 7).

Finally, to evaluate the impact of PLCγ1 activation on VEGF-A-induced vascular responses in vivo, we used mouse corneal micropocket assay to examine the effect of selective PLCγ inhibitor U73122. We observed a dramatic inhibition of VEGF-

**Fig. 5 NDRG1 deficiency impairs VEGF-A signaling by decreasing PLCγ1 activation. a** Schematic image of late process of VEGF-A signaling, PLCγ1 activation, and the enhanced expression of IP$_3$, DAG, and Ca$^{2+}$ in cytoplasm, resulting in ERK1/2 activation. **b** Western blot analysis shows phosphorylation of PLCγ1, VEGFR2, and ERK1/2 in WT- and KO-mECs when treated with 50 ng/ml VEGF-A for indicated time. We normalized phosphorylated PLCγ1 in the presence of VEGF-A to that at 0 min (1.0) ($n = 3$ per group). **c** Western blot analysis shows phosphorylation of PLCγ1, VEGFR2, and ERK1/2 in HUVECs following siRNA-mediated silencing of NDRG1 when treated with 20 ng/ml VEGF-A for indicated time. We normalized phosphorylated PLCγ1 in the presence of VEGF-A to that at 0 min (1.0) ($n = 3$ per group). **d** Western blot analyses show protein expression levels of PLCγ1, VEGFR2, ERK1/2, and AKT in HUVECs when treated with 20 ng/ml VEGF-A or 10 ng/ml FGF-2 for indicated times. We normalized phosphorylated PLCγ1 in the presence of VEGF-A or FGF-2 to that at 0 min (1.0) ($n = 3$ per group). **e** We present the time course of mean Fluo-4 fluorescence intensity/cell. We stimulated HUVECs with VEGF (50 ng/ml) or FGF-2 (20 ng/ml) for the indicated time ($n = 3$ per group). **f** The effects of the Ca$^{2+}$ selective chelator BAPTA-AM on VEGF-A-induced cellular calcium levels; we treated HUVECs with or without 50 μM BAPTA-AM for 2 h and then stimulated them with VEGF (50 ng/ml) for indicated times ($n = 3$ per group). **g** Effects of BAPTA-AM on phosphorylation of VEGFR2, PLCγ1, PKCδ, and ERK1/2. We normalized phosphorylated ERK1/2 to VEGF-A-unstimulated cells (1.0) ($n = 3$ per group). **h** Effects of the PKC activator PMA on the expression and phosphorylation of PLCγ1, PKCδ, and ERK1/2. We normalized phosphorylated ERK1/2 to PMA-unstimulated cells (1.0) ($n = 3$ per group). We present data as means ± SD of $n$ observations, and we identified differences using two-tailed $t$-test, *$P < 0.05$, **$P < 0.01$. **b–d**, **g**, **h** Source data are provided as Supplementary Fig. 3.

A-induced neovascularization in the corneal micropocket assay when treated with the PLCγ inhibitor (Supplementary Fig. 5a). Whole-mount immunostaining revealed a reversal of the development of CD31+ blood vessels on day 7 after the implantation of pellets containing VEGF-A in the U73122-treated cornea (Supplementary Fig. 5b). VEGF-A-induced angiogenesis was significantly (**$P < 0.01$) suppressed by the PLCγ inhibitor in cornea (Supplementary Fig. 5c). Together, these results demonstrate the essential role played by NDRG1-induced PLCγ1 activation in VEGF-A-induced angiogenesis.

## Discussion

Our previous study demonstrated that, due to decreased infiltration of macrophages in the angiogenic areas, *NDRG1* deficiency induced impaired angiogenesis in response to inflammatory cytokines[18]. In this study, to our knowledge, we demonstrate the novel finding that NDRG1 deficiency markedly attenuated VEGF-A-induced angiogenesis, further noting that NDRG1 is abundantly expressed in ECs in human tumor samples (Fig. 1a), and hypothesizing that NDRG1 function in ECs is indispensable for VEGF-A-induced angiogenesis. Finally, we find that, through its close association with PLCγ1, NDRG1 acts as a key positive regulator of VEGF-A-induced angiogenesis. Furthermore, we also suggest that NDRG1-mediated PLCγ1 activation might be a reliable therapeutic target for VEGF-A-mediated vascular diseases, including cancer.

Among the 13 mammalian PLC isozymes, PLCγ1 belongs to the PLCγ subtype (γ1 and γ2), which is directly activated by receptor tyrosine kinases[33]. While PLCγ1 is ubiquitously expressed in multiple tissues, PLCγ2 is mainly expressed in hematopoietic cell lineages[34]. In our present study, selective PLCγ1 inhibitor U73122 markedly suppressed VEGF-A-induced neovascularization in mouse cornea (Supplementary Fig. 5). Consistent with our data, a previous study reported the inhibition of PLCγ1 activation or expression to almost completely block VEGF-A-induced angiogenesis by Matrigel plug assay[35]. Our data also suggest that PLCγ1 activation is indispensable for VEGF-A-induced angiogenesis. PLCγ1 thus, through enhanced expression of IP$_3$ and DAG, Ca$^{2+}$ release from ER to cytoplasm, and activation of ERK1/2, plays a key role in VEGF-A-induced angiogenesis (Fig. 7).

Here, first, we identified that PLCγ1 forms a complex with NDRG1 in ECs irrespective of the presence or absence of VEGF-A, and that the phosphorylation sites of NDRG1 are important for the interaction with PLCγ1. Presently, the requirement of NDRG1 phosphorylation for the formation of an NDRG1 and PLCγ1 complex remains unclear. In addition, in hECs the expression of PLCγ1 and NDRG1 was increased in the cell membrane fraction after stimulation with VEGF-A. Together,

these results suggest the likelihood that NDRG1 activates PLCγ1 by their mutual interaction, and coordinates the trafficking of PLCγ1 to the cell membrane, supporting the gear-up of VEGF-A signaling (Fig. 7). On the other hand, FGF-2 did induce ERK1/2 activation without PLCγ1 activation in ECs. PLCγ1 activation is thus required for VEGF-A-induced angiogenesis, but not FGF-2-induced angiogenesis. Further, concerning of the essential role of PLCγ1 in various cell signaling pathways and cellular events, our findings will further facilitate to understand the mechanisms of PLCγ1-related biological events under physiological and pathological condition.

It is widely known that the function of NDRG1 is closely associated with growth, differentiation, and development[13]. However, *Ndrg1* deficiency is not lethal to embryos[18,36]. We observed that *Ndrg1* deficiency retarded the development of retinal blood vessels, but did not affect the electrophysiological function of the retina (unpublished data). Thus, *Ndrg1* deficiency only partially suppresses the development of retinal vessels, if it does so at all. In contrast, deletion mutation of the *Vegfa* or *Vegfr2* gene is lethal to the embryo due to abnormal vascular development[37,38]; deletion mutation of the *Plcg1* gene induces embryonic death after day 8.5, and embryos show impaired vasculogenesis[39]. Regarding the non-embryonic lethality by NDRG1 deficiency, further study should clarify whether other NDRG family members, such as NDRG2, NDRG3, and NDRG4 or other factors could compensate for the absence of the NDRG1 function in the development of neovessels.

In conclusion, to our knowledge, we presented here a novel mechanism of angiogenesis in which NDRG1 in ECs acts as a key effector in VEGF-A-induced angiogenesis during the late VEGF-A signaling process to activate the PLCγ1/ERK1/2 signaling pathway. The present evidence of NDRG1-mediated VEGF-A signaling could be exploited in the design of specific therapeutic drugs for VEGF-A-mediated vascular diseases, including cancer.

## Methods

**Reagents and antibodies.** Polyclonal antibody against full-length Ndrg1 (1:1000) was a kind gift from Dr. Kokame[40] (National Cerebral and Cardiovascular Center, Suita, Japan). We purchased anti-mouse VEGFR-2 antibody (1:25, ab10972) from Abcam (Cambridge, UK), and CF488 conjugated anti-goat IgG antibody (1:25, 20225) from Biotium (Hayward, CA), and Alexa Fluor® 488 anti-mouse CD102 (ICAM-2) antibody (1:200, 105609) and PE anti-mouse CD31 antibody (1:20, 102407) from BioLegend (San Diego, CA), FITC-conjugated Mouse IgG2b, κ Isotype Control (1:200, 559532), and PE-conjugated Mouse IgG2a, κ Isotype Control (1:20, 553457) from Becton-Dickinson (San Diego, CA) for flow cytometry analysis. We generated anti-NDRG1 antibody for Co-IP assay as described previously[41] (1:500). We purchased mouse anti-PLCγ1 antibody (1:100, sc-7290) from Santa Cruz Biotechnology (Santa Cruz, CA), and rabbit anti-NDRG1 antibody (1:100, ab124689) from Abcam for PLA assay. We purchased the following antibodies from Cell Signaling Technology (Beverly, MA): anti-phospho-ERK1/2 (T202/Y204) (1:1000, 9106); anti-ERK1/2 (1:1000, 9102); anti-phospho-AKT (S473) (1:2000, 4060); anti-AKT (1:1000, 9272); anti-NDRG1 (1:1000, 9408); anti-

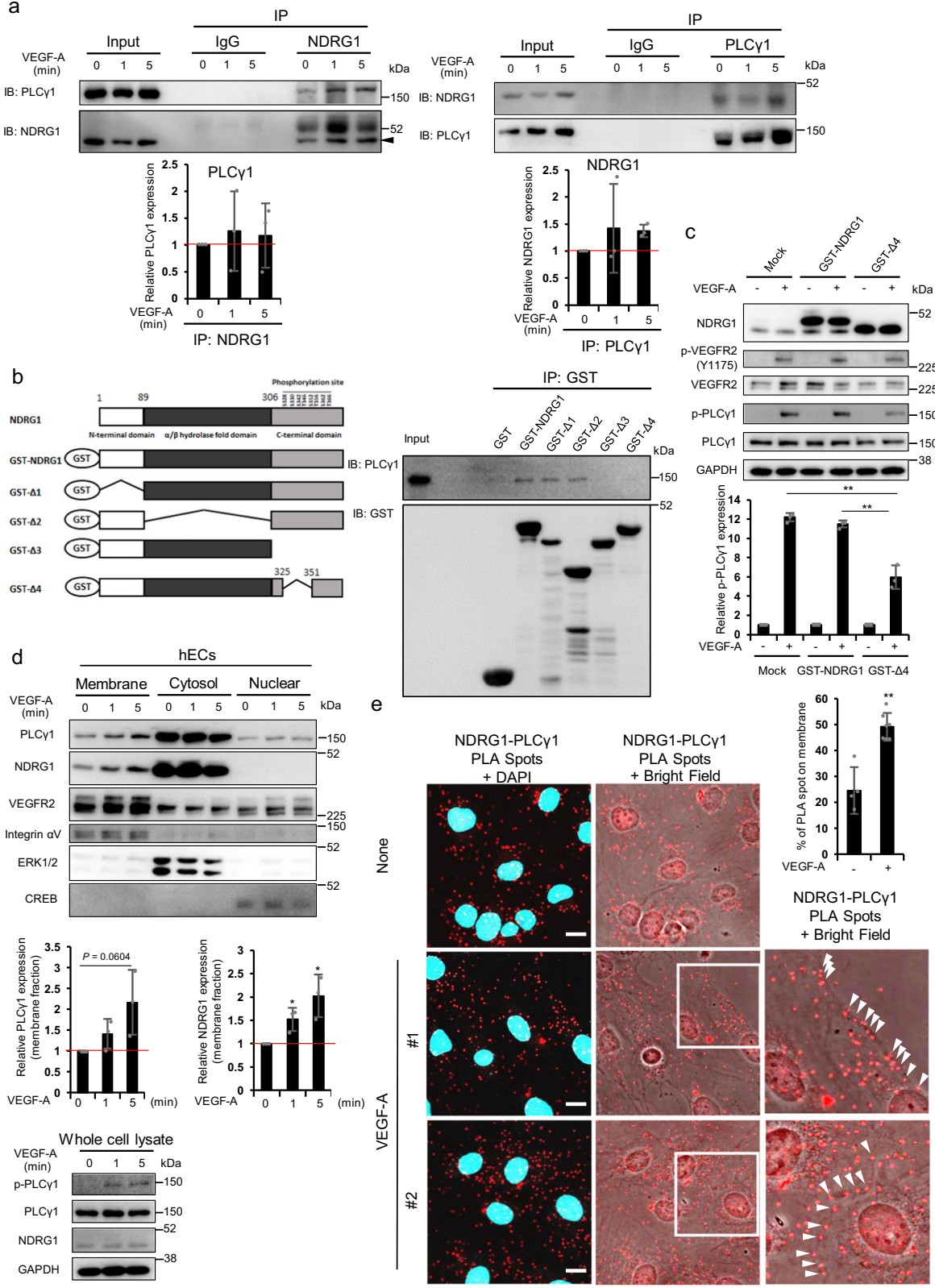

FGF receptor (FGFR)1 (1:1000, 9740); anti-phospho-PLCγ1 (Y783) (1:1000, 14008); anti-PLCγ1 (1:1000, 5690); anti-phospho-VEGFR2 (Y1175) (1:1000, 2478); anti-VEGFR2 (1:1000, 2479); anti-phospho-PKCδ (T505) (1:1000, 9374); and anti-PKCδ (1:1000, 9616). We purchased the anti-GAPDH antibody (1:5000, 2275-PC-100) from Trevigen (Gaithersburg, MD), the recombinant human VEGF 165 protein, CF (293-VE/CF) and recombinant human FGF basic (146 aa) protein, CF (233FB/CF) from R&D system Inc. (Minneapolis, MN), the BAPTA-AM (03731-24) from Nacalai Tesque Inc. (Kyoto, Japan), and the PLC-γ inhibitor U73122

(1268) from Tocris (Abingdon, UK). Primary antibodies were validated for use based on the position the antigen in SDS-PAGE gels. We also used the NDRG1 KO or siRNAs to specifically deplete endogenous protein to verify anti-NDRG1 antibodies. All other commercial antibodies were validated by the manufacturers.

**Immunofluorescence analysis**. We analyzed five patients newly diagnosed with breast cancer, who were treated between 2017 and 2018 at Kurume University

**Fig. 6 NDRG1 specifically interacted with PLCγ1 through its phosphorylation sites, and translocated into cell membrane by VEGF-A. a** Association of PLCγ1 with NDRG1 in the absence or presence of VEGF-A. We subjected immunoprecipitates to western blot analyses using indicated antibodies against NDRG1 or PLCγ1. We normalized expression levels of PLCγ1 and NDRG1 in the presence of VEGF-A to that at 0 min (1.0) (n = 3 per group). **b** We incubated immobilized tagged proteins with total lysate of HUVECs, and analyzed the bound proteins using western blotting with an anti-PLCγ1 antibody. **c** Western blot analysis in HUVECs upon transfection with Mock, GST-NDRG1, or GST-Δ4 cDNA when treated with VEGF-A for 5 min. Quantitative analysis of phosphorylated PLCγ1 expression with or without VEGF-A for 5 min. PLCγ1 phosphorylation levels in the presence of VEGF-A were normalized to those in unstimulated cells (1.0) (n = 3 per group). **d** We determined subcellular localization of PLCγ1 and NDRG1 by western blot using membrane, cytoplasmic, and nuclear fractions of HUVECs treated with VEGF-A for indicated time. We used integrin αV, ERK1/2, and CREB as loading controls for the membrane, cytoplasmic, and soluble nuclear fractions, respectively. Relative expression levels in membrane fraction in the presence of VEGF-A were normalized to those in unstimulated cells (1.0) (n = 3 per group). **e** PLA to determine the interaction of NDRG1 and PLCγ1 in HUVECs. A red fluorescent spot indicates an existence of a proximal localization and/or an interaction of NDRG1 and PLCγ1. In the enlarged insets, we have superimposed bright-field images onto the fluorescence images to reveal the cell borders. Arrowheads indicate PLA spots around cell membrane. The percentage of PLA spots on cell membrane in the absence and presence of VEGF-A (None: n = 4, VEGF-A: n = 7). A positive and a negative control were shown in Supplementary Fig. 6. Scale bar = 10 μm. We present data as means ± SD of n observations, and we identified differences using two-tailed t-test, *P < 0.05, **P < 0.01. **a–d** Source data are provided as Supplementary Fig. 3.

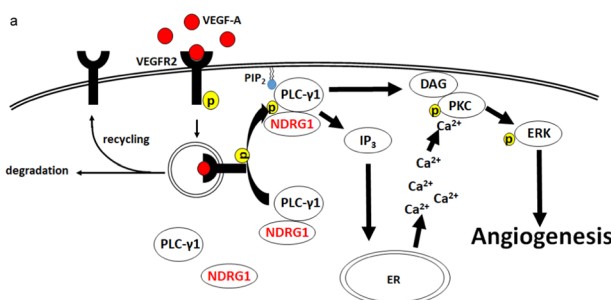

**Fig. 7 NDRG1 promotes VEGF-A-induced angiogenesis through PLCγ1/ERK signaling in vascular endothelial cells. a** A model of the mechanisms by which NDRG1 promotes VEGF-A-induced angiogenesis via the VEGFR2/PLCγ1 pathway in vascular endothelial cells. IP$_3$ inositol 1,4,5-trisphosphate, DAG diacylglycerol, PIP$_2$ phosphatidylinositol 4,5-bisphosphate.

Hospital (Kurume, Japan). We accomplished immunofluorescence multiplex staining with PerkinElmer Opal kit (Perkin-Elmer, Waltham, MA). For immunofluorescence analysis of the tumors of breast cancer patients, we cut each formalin-fixed paraffin embedded tissue sample into 4 μm slices and examined it on a coated glass slide. We performed antigen retrieval in citrate buffer (pH 6.0) using microwave treatment. We washed slides and performed blocking with 3% H$_2$O$_2$ blocking solution. We incubated the first primary antibodies for anti-NDRG1 (1:200, ab124689) for 30 min in a humidified chamber at room temperature and detected them using the Opal™ Polymer HRP Ms+Rb kit (ARH1001EA; PerkinElmer, Waltham, MA). We accomplished visualization of NDRG1 using Opal 520 fluorophore (1:100), after which we placed the slide in citrate buffer (pH 6.0) and heated it using microwave treatment. In a serial fashion, we then incubated the slide with primary antibodies for anti-CD34 (1:100, PA0212; Leica Biosystems) for 30 min in a humidified chamber at room temperature, and detected them using the Opal™ Polymer HRP Ms+Rb kit. We then visualized the CD34 using Opal 570 fluorophore (1:100), and placed the slide in citrate buffer (pH 6.0) for microwave treatment. We subsequently visualized the nuclei with DAPI.

**Mice**. We purchased the Ndrg1$^{-/-}$ mice on a C57BL6 background from Laboratory Animal Resource Bank, National Institutes of Biomedical Innovation, Health and Nutrition (Osaka, Japan). In all our experiments we compared Ndrg1$^{-/-}$ mice with age- (6–10 weeks) and gender-matched Ndrg1$^{+/+}$ mice[18]. We housed the mice in a specific-pathogen free room. We housed all mice in microisolator cages maintained under a 12 h light/dark cycle, and supplied them with water and food ad libitum. We observed animals for signs in accordance with the guidelines of the Harvard Medical Area Standing Committee on Animals.

**PCR analysis**. We performed PCR analysis of DNA isolated from cut tail as described previously[18]. PCR analysis of DNA isolated from cut tail with primers: P1: AGCAGGCTCTTAAAGCGGCTCC, P2: CCGCCTCTGTCAAATTAGTAG CTG, and P3: GGGAGAGCTGAAGGCTGTTCTAGG. The product of P2 + P3 gives a wild-type band of 201 bp. A Ndrg1$^{-/-}$ band of 268 bp is produced by primers P1 + P3. We performed PCR in a final volume of 20 μl using Takara Ex Taq (RP001B; Takara Bio, Shiga, Japan). We performed initial denaturing at 94 °C for 2 min followed by 35 cycles (94 °C for 30 s, 53 °C for 30 s, 72 °C for 2 min) and a final extension at 72 °C for 5 min.

**Cell culture**. We obtained murine RENCA cells from the American Type Culture Collection (Manassas, VA). We cultured cancer cells in RPMI-1640 medium (Nissui, Tokyo, Japan) supplemented with 10% fetal bovine serum (FBS) (Hyclone, Logan, UT). We cultured mouse lung ECs in endothelial cell basal medium (EBM-2) supplemented with the EGM-2MV Bullet Kit (15% FBS) (CC-3202; Lonza Walkersville Inc., Walkersville, MD). We purchased HUVECs from Lonza Walkersville Inc. and cultured them in EBM-2 supplemented with the EGM-2 Bullet Kit (2% FBS) (CC-3162; Lonza Walkersville Inc). We passaged all cell lines for ≤6 months and they were not further tested or authenticated by the authors. None of the cell lines used in this study was authenticated by the authors. For primary mouse lung ECs, we checked the expression of vascular endothelial markers CD31, ICAM-2, VEGFR2, and VE-cadherin. All cell lines were routinely tested negative for mycoplasma contamination. No commonly misidentified cell lines were used.

**Mouse dorsal air sac assay**. We carried out the assay in male mice at 6–10 weeks of age, as described previously[25]. We suspended RENCA cells (2 × 10$^6$) in 150 μl of phosphate-buffered saline (PBS) and injected them into a chamber comprising a ring covered with Millipore filters (0.45 μm pore size) (P1688; Merck Millipore, Darmstadt, Germany) on each side. On day 5, we removed the chambers from the subcutaneous fascia and replaced them with black rings with the same inner diameter as the chambers. We assessed photographs of these sites by counting the number of newly formed vessels.

**Corneal micropocket assay**. We performed the corneal micropocket assay essentially as described previously[18]. Briefly, we prepared 0.3 μl of Hydron pellets (P3932; Sigma-Aldrich, St-Louis, MO) containing human VEGF-A (200 ng) or FGF-2 (100 ng) and implanted them in the corneas of mice (6–10 weeks old). To examine the effect of PLCγ inhibition we applied a PLCγ inhibitor U73122 (30 μM, 2 μl eye drops) topically to VEGF-A with 30 μM U73122-containing pellets implanted in the eyes once a day from day 1 to day 6. After day 7, we sacrificed the mice and photographed their corneal vessels. We analyzed areas of corneal neovascularization using ImageJ and expressed them in mm$^2$.

**Whole-mount immunofluorescence**. We enucleated the mouse eyes and fixed them with 4% paraformaldehyde for 30 min at 4 °C. For whole-mount preparation, we microsurgically exposed the corneas by removing other portions of the eye. Then, we made radial cuts into the cornea. We washed the tissues with PBS, three times for 5 min, and, then, placed them in methanol for 20 min. We incubated the tissues overnight at 4 °C with anti-mouse CD31 mAb (5 mg/ml, 550274; BD Biosciences, San Diego, CA, USA) diluted in PBS containing 10% goat serum and 1% Triton X-100. We then washed the tissues four times for 20 min in PBS and incubated them with CF594-conjugated anti-rabbit IgG antibody (1:500, 20152-1; Biotium) overnight at 4 °C. We prepared corneal flatmounts on glass slides using a mounting medium (TA-030-FM; Lab Vision Corporation). We performed fluorescence imaging with KEYENCE BZ-8000.

**Aortic ring assay**. We anesthetized mice (6–10 weeks old) and harvested the aorta. We embedded 1 mm mouse aortic rings in three-dimensional growth factor reduced Matrigel (356231; Corning Inc., Corning, NY), treated with or without VEGF-A (25 ng/ml) or FGF-2 (50 ng/ml), and incubated them at 37 °C. We measured the vascular length and branching point at day 7.

**Isolation of mouse lung ECs**. We isolated CD31+ ECs from mouse lung by magnetic sorting using CD31 MicroBeads (1:10, 130-097-418; Miltenyi Biotec GmbH, Bergisch-Gladbach, Germany). Briefly, we anesthetized the mice (6–10 weeks old) and harvested lung tissue samples. We incubated lung tissues minced in PBS in collagenase A (10103586001; Roche Diagnostics GmbH,

Mannheim, Germany) and deoxyribonuclease I (LS002138; Worthington Biochemical, Lakewood, NJ) at final concentrations of 0.5% and 20 unit/ml. We incubated the mixture for 1 h at 37 °C under gentle agitation. We stopped digestion using FBS, after which we washed the cell suspension and then passed it through a 100-μm mesh nylon screen. We incubated the cells with CD31 MicroBeads for 15 min at 4 °C and loaded them onto a MidiMACS kit (LS) (130-042-301; Miltenyi Biotec GmbH) according to the manufacturer's instructions.

**Endothelial cell-proliferation assay**. We seeded ECs ($2.5 \times 10^4$ cells/well) in type I collagen (08-115; Merck Millipore)-coated 24-well plates. After 24 h, we incubated 0.2% serum-starved ECs for another 24 h, and cultured them for 48 h in the presence or absence of VEGF-A (20 ng/ml) or FGF-2 (10 ng/ml). We harvested and counted the cells on a Coulter counter (Beckman Coulter Inc., Brea, CA).

**Endothelial cell-migration assay**. We performed the migration assay using a multiwell chamber as the outer chamber and used as the inner chamber (CH8–24; Kurabo, Osaka, Japan) with 8-μm pore polycarbonate filters coated with 1.33 g/ml fibronectin (33016-015; Thermo Fisher Scientific, Waltham, MA), as described previously. Briefly, we seeded ECs ($1.0 \times 10^5$ cells/well) in EBM-2 containing 1% FBS in the inner chamber, and added VEGF-A (50 ng/ml) or FGF-2 (20 ng/ml) to the same medium in the outer chamber. After 6 h incubation, we counted the cells that had migrated under the filter manually by microscopic examination.

**Western blot analysis**. We rinsed the cells with ice-cold PBS and lysed them in buffer containing 50 mM of Tris-HCl, 350 mM of NaCl, 0.1% Nonidet P-40, 5 mM of EDTA, 50 mM of NaF, 1 mM phenylmethylsulfonyl fluoride, 10 μg/ml of leupeptin, and 1 mM of $Na_3VO_4$. We subjected cell lysates to western blotting as described previously[42]. We quantified the intensity of luminescence using a charge-coupled device camera combined with an image-analysis system (LAS-4000; GE Healthcare Life Science, Little Chalfont, UK). We analyzed the data using ImageJ.

**RNA interference**. We purchased three duplex siRNAs that specifically target different regions of human NDRG1 mRNA (HSS173605) from Invitrogen (Carlsbad, CA). We transfected the HUVECs with siRNA duplexes using Lipofectamine RNAiMAX (13778150; Invitrogen) and Opti-MEM (31985070; Invitrogen) according to the manufacturer's recommendations. Forty-eight hours later, we starved the cells with 0.2% serum for 24 h.

**Flow cytometry**. We detached mouse lung ECs with 5 mM EDTA and washed them in PBS, prior to flow cytometry analysis. We stained mouse lung ECs in suspension containing $1 \times 10^6$ cells/100 μl in incubation buffer (0.5% BSA in PBS) with anti-VEGFR2 mAb or FITC-conjugated Mouse IgG1 Isotype Control (555748; Becton-Dickinson) for 25 min at 4 °C, and thereafter incubated them in CF488-conjugated anti-goat IgG antibody for 20 min at 4 °C. For CD31 and ICAM-2 staining, we stained mouse lung ECs in suspension containing $1 \times 10^6$ cells/100 μl in incubation buffer (0.5% BSA in PBS) with Alexa Fluor® 488 anti-mouse ICAM-2 antibody (1:200; BioLegend) and PE anti-mouse CD31 antibody (1:20; BioLegend) or FITC-conjugated Mouse IgG2b, κ Isotype Control (1:200, 559532; Becton-Dickinson) and PE-conjugated Mouse IgG2a, κ Isotype Control (1:20, 553457; Becton-Dickinson) for 25 min at 4 °C. We used a FACSCalibur system (Becton-Dickinson) to acquire data. We used the CellQuest Pro software program (BD Biosciences, San Jose, CA) to analyze the data.

**VEGF-A binding assay**. We used human VEGF biotinylated fluorokine flow cytometry kit (NFVE0; R&D Systems) for this binding assay, which we performed according to the manufacturer's instructions. We added a total of 10 μl biotinylated VEGF reagent to 25 μl of ECs ($4 \times 10^6$ cells/ml). As a negative control, we stained an identical sample of cells with 10 μl biotinylated negative control reagent. We incubated the cells for 60 min on ice. Then, we added 10 μl avidin–FITC reagent to each sample, and incubated the reaction mixture for an additional 30 min on ice. After incubation, we washed the cells twice with 2 ml of 1× cell wash buffer to remove unreacted avidin–FITC, and then resuspended the cells in 0.2 ml of 1× cell wash buffer for flow cytometric analysis.

**VEGFR2 internalization assay**. We grew mECs or hECs to confluence on 60 mm dishes and starved them overnight in media with 0% or 0.2% FBS. We incubated the cells with 0.2 mg/ml EZ-Link Sulfo-NHS-Biotin (Thermo Scientific, 10509863) in PBS on ice for 30 min, quenched with cold PBS + 50 mM glycine. Then, we stimulated the cells with VEGF-A (50 or 20 ng/ml) at 37 °C. At predetermined time points, we rinsed cells with cold PBS and incubated them for 15 min on ice with 225 mM sodium 2-mercaptoethanesulfonate (Tokyo Chemical Industry, M0913) to cleaved biotin moieties remaining at the cell surface. After 15 min, we incubated the cells for 10 min on ice with 900 mM iodoacetamide (Nacalai Tesque Inc., 19302-54) added. We rinsed the cells with cold PBS and protein lysates prepared using lysis buffer (50 mM of Tris-HCl, 350 mM of NaCl, 0.1% Nonidet P-40, 5 mM of EDTA, 50 mM of NaF, 1 mM phenylmethylsulfonyl fluoride, 10 μg/ml of leupeptin, and 1 mM of $Na_3VO_4$). We immunoprecipitated lysate with Strepdavidin

Agarose beads (Thermo Fisher Scientific, 20349) at 4 °C overnight, and measured the internalized VEGFR2 fraction by western blotting with antibodies to VEGFR2.

**VEGFR2 recycling assay**. We starved HUVECs with 0.2% serum overnight and then stimulated them with VEGF-A (50 ng/ml) for 30 min in the presence of cycloheximide (10 μM). For the assessment of VEGFR2 recycling, we additionally washed the cells three times with warm PBS and incubated them for 1 h in 0.2% serum medium in the presence of cycloheximide (10 μM). Next, we put the cells on ice, washed them with ice-cold PBS and, then, incubated them with 0.5 mg/ml EZ-Link Sulfo-NHS-Biotin (Thermo Scientific, 10509863) dissolved in PBS for 1 h to biotinylate surface receptors. We stopped the reaction with ice-cold PBS containing 50 mM glycine. Finally, we collected the cells in ice-cold lysis buffer (50 mM of Tris-HCl, 350 mM of NaCl, 0.1% Nonidet P-40, 5 mM of EDTA, 50 mM of NaF, 1 mM phenylmethylsulfonyl fluoride, 10 μg/ml of leupeptin, and 1 mM of $Na_3VO_4$). After centrifugation, we incubated the supernatant with Strepdavidin Agarose beads (Thermo Fisher Scientific, 20349) overnight on a rotator at 4 °C to precipitate biotinylated proteins. We washed the beads with lysis buffer and eluted the biotinylated proteins in 40 μl of 2× SDS sample buffer, boiled them for 10 min, and subjected them to western blot analysis.

**Intracellular $Ca^{2+}$ measurements**. We used Calcium Kit-Fluo 4 (CS22; Dojindo, Kumamoto, Japan) for this assay, which we performed according to the manufacturer's instructions. We loaded serum-starved ECs for 1 h at 37 °C with Fluo-4-AM in a loading buffer, according to the manufacturer instructions. We replaced the loading buffer with pre-warmed recording buffer. Subsequently, we stimulated the cells with 50 ng/ml VEGF and recorded the levels of intracellular $Ca^{2+}$ for 20 min (1 min time lapses). We measured absorbance at 485 nm using a 96-well plate reader (DTX800; Beckman Coulter Inc.).

**Co-IP assay**. We rinsed cells in ice-cold PBS and lysed them in IP lysis buffer (50 mM Tris-HCl (pH 8.0), 250 mM NaCl, 1 mM EDTA, 10% glycerol, 0.3% NP-40). We incubated the extract overnight at 4 °C with antibodies to NDRG1 (1:400) or PLCγ1 (1:50) in NET-gel buffer (50 mM Tris-HCl (pH 7.5), 150 mM NaCl, 1 mM EDTA, 0.25% gelatin, 0.02% sodium azide, 1 mM PMSF, 10 mg/ml leupeptin, and 10 mg/ml aprotinin), and, then, with Protein G Agarose (16-266; Merck Millipore) for 1 h at 4 °C. We washed the beads three times with NET-gel buffer. After centrifugation, we boiled the precipitate and starting material in western sample buffer for SDS-PAGE and western blot analysis.

**Pulldown assay**. We performed the pulldown assay as described previously[42]. Briefly, preparation of the GST-tagged NDRG1 and NDRG1 deletion mutants, Δ1–Δ4, expression plasmids was as described previously[10]. We dialyzed GST fusion proteins against X-buffer (50 mM; Tris-HCl, pH 8.0; 1 mM EDTA; 120 mM NaCl; 0.5% NP-40 10% glycerol; and 1 mM PMSF). We incubated immobilized GST or GST fusion proteins with Glutathione Sepharose 4B (GE Healthcare) for 4 h at 4 °C. After five washes with X-buffer (1 ml), we incubated GST fusion proteins with total cell lysate of HUVECs overnight at 4 °C. We washed the complex five times with X-buffer and, then, subjected it to SDS-PAGE.

**Overexpression of NDRG1 or NDRG1 mutants**. GST-tagged NDRG1, GST-tagged NDRG1 deletion mutants, Δ4, or vector was transfected by electroporation using Amaxa™ HUVEC Nucleofector Kit (VPB-1002, Lonza). After transfection, HUVECs ($3.0 \times 10^5$ cells) were seeded in 35 mm dishes. The following day, we starved HUVECs with 0.2% serum overnight and then stimulated them with VEGF-A (20 ng/ml) for 5 min.

**Subcellular protein fractionation**. We performed cellular fractionation using a Thermo Scientific, Subcellular Protein Fractionation Kit for Cultured Cells (78840), according to the manufacturer's instructions.

**Proximity ligation assay**. We performed the PLA using Duolink PLA In Situ Red Starter Kit Mouse/Rabbit (DUO92101; Sigma-Aldrich) according to the manufacturer's instructions. We used mouse anti-PLCγ1 antibody (1:100) and rabbit anti-NDRG1 antibody (1:100) as primary antibodies. We acquired images using a Fluoview FV10i (Olympus).

**Study approval**. The Animal Ethics Committee of Kyushu University (Fukuoka, Japan) reviewed and approved all the animal experimental procedures in this study (approval numbers: A28-029-1 and A30-068-0). The study using human tumor tissue samples conforms to the principles of the Declaration of Helsinki and the need for written informed consent was waived for this retrospective study. This study was approved by the Institutional Review Board of Kurume University Hospital (Kurume, Japan) (approval number: 18235).

**Statistics and reproducibility**. We express all results as the mean ± standard errors of the mean (SE) (Figs. 1c–e, 2b, Supplementary Fig. 5c) or standard deviation (SD) (Figs. 1b, 2c, d, 3a–d, 5b–h, 6a, 6c, e) of $n$ observations, and assessed

statistical differences among the groups by one- (Figs. 2c, d, 3c) or two-tailed (Figs. 1b–e, 3a, b, 3d, 5b–h, 6c–e, Supplementary Fig. 5c) Student's *t*-test. We considered a *P* value of less than 0.05 to be significant. All attempts at replication were successful. Each individual experiment was repeated independently with similar results at least three times. The sample size was determined based on the results from pilot studies and previous experience of the variability of each data within control groups.

**Reporting summary**. Further information on research design is available in the Nature Research Reporting Summary.

## Data availability
All data supporting the conclusions are available from the authors on reasonable request. The source data underlying Figs. 1b–e, 2b–d, 3a–d, 5b–h, 6a, c–e are provided as Supplementary Data 1. Full blots are shown in Supplementary Fig. 3.

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

## Acknowledgements
This research was supported by JSPS KAKENHI grant numbers JP25830083, JP15K18411, JP17K07169 (to K.W.), and JP17H06349 (to M.O.), the Life Science Foundation of Japan (to K.W.), the Takeda Science Foundation (to K.W.), and St. Mary's Institute of Health Sciences (to K.W., M.K., and M.O.).

## Author contributions
K.W., T.S., H.F., and A.S. performed most of the experiments, analyzed data with assistance from Y.M., H.I., and S.Y., H.A., A.K., and J.A. performed co-immunofluorescent analysis; K.W., M.K., and M.O. designed the experiments, wrote the manuscript with help from T.S. and A.S.; all authors reviewed and commented on the manuscript.

## Competing interests

The authors declare no competing interests.
