## [Peer Review File · Communications Biology]

Reviewers' comments:

Reviewer #1 (Remarks to the Author):

Brief summary of the manuscript:

In this manuscript, the authors aimed to investigate the novel role of NDRG1 in VEGF A-induced angiogenesis and signaling in endothelial cells. Previous work by the same laboratory showed that NDRG1 plays a critical role in VEGF expression in macrophages and in tumor cells (Scientific Reports 2016, Cancer Research 2009). In this study, the authors used wild type and NDRG-1 ko mice in a series of in vivo angiogenesis models, such as the cornea micropocket angiogenesis assay or the mouse dorsal air sac assay of tumor angiogenesis, as well as ex vivo, such as the angiogenesis aortic ring model. In addition, they conducted in vitro functional and signaling studies with lung endothelial cells isolated from wild type and NDGR-1 ko mice and HUVEC using siRNA to knock down NDRG-1 to elucidate the role of NDGR-1 in VEGF A signaling.

The authors found that NDRG-1 plays a critical role in VEGF-induced, and not FGF-induced, angiogenesis in vivo. They also report that NDRG1 is primarily detected in the endothelium in human tumor biopsies. In vitro, NDRG-1 is required for PLC-gamma and ERK1/2 activation downstream of VEGFR2.

Overall impression of the work:

Overall strengths: The in vivo and in vitro functional studies convincingly show the critical role of NDGR-1 in tumor/VEGF-induced angiogenesis. Quantitative data are provided and the studies are very comprehensive. With regards to the in vitro signaling data, the authors used primary mouse lung endothelial cells isolated from wild type and NDRG1 knock out mice as well as human umbilical vein endothelial cells and siRNA approaches to acutely downregulate NDRG-1.

Overall weaknesses: the purity of the endothelial cells isolated from mouse lungs is unclear. Lack of quantification of many of the in vitro signaling studies.

Specific comments, with recommendations for addressing each comment:

The following are the major issues that need to be addressed by the authors:

1. There is no data available regarding the purity of the mouse lung endothelial cells as well as the cell passages used for the experiments. The authors indicated that endothelial cells were sorted with a CD31 antibody. However, in order to get pure endothelial cell populations it is required to conduct a second sorting, with an ICAM-2 antibody at the time of the first trypsinization when cells reach confluence.
2. In figure 5, it is not clear that NDGR-1 is required for PLC gamma phosphorylation. Quantitative data should be provided.
3. In figure 6, the data showing recruitment of NDGR-1 and PLC-gamma to the same complex is not convincing. Additional experiments should be conducted and quantitative data should be provided.
4. The authors do not provide any data on how the NDRG-1 antibody used for human samples was validated.
5. In page 15, lines 231-240, the data should be described more clearly.

Reviewer #2 (Remarks to the Author):

In this manuscript, Watari and co-workers demonstrated that N-myc downstream regulated gene 1 (NDRG1) is playing a significant role in VEGF-A, a pro-angiogenesis growth factor, mediated in angiogenesis process. Using *NdrG1*^{-/-} mice, the authors showed that lack of NDRG-1 expression impaired VEGF-A-induced angiogenesis especially in corneas. In addition, the authors also demonstrated that tumor-induced angiogenesis by a highly-expressing VEGF-A cancer cell lines was reduced significantly when implanted in a *NdrG1*^{-/-} mice dorsal air sac as compared to that of control mice. In a different model, the authors also showed that the EC lack of NDRG1 exhibited less angiogenic sprouting from the aorta and NDRG1 is in the similar pathways for the activation of phospholipase C γ 1 (PLC γ 1) and ERK1/2 by VEGF-A without influencing the expression and function of VEGF receptor-2. It was also been shown that NDRG1 formed a immunocomplex with PLC γ 1 through its phosphorylation sites, and the inhibition of PLC γ 1 dramatically suppressed VEGF-A-induced angiogenesis in the mouse cornea, strongly suggesting the essential role of NDRG1 in VEGF-A-induced angiogenesis through PLC γ 1 signaling. There are several interesting observations in the manuscript, however there are some questions/concerns need to be addressed to support the authors' hypothesis. Comments:

- a) One major problem of this manuscript is lack of developing the logistics of the experimental presentations and the Figures. For examples in Figure 1, the authors showed that RENCA cells which is highly expressing VEGF-A when implanted in mouse dorsal ear sac assays. But they didn't try to grow the cells for tumor growth, which is highly VEGF induced angiogenesis dependent. There animal model is the best model to test the hypothesis.
- b) Similarly, Fig 2 started by showing that in breast cancer patients tissues expresses NDRG1 (by CD34 co-localization) in EC and then they started aortic ring experiments to show that role of NDRG1 in VEGF induced cell proliferation and migration in EC. Very hard to understand the logistic.
- c) Does VEGF treatment increase NDRG1 protein level within 7 min treatment? It is not clear why that was happened, as per Fig 3a. Which one is the real band of NDRG1? It is surprising that the authors didn't try to show the phosphorylation of tyrosine specific VEGFR2 in those experiments. In Fig 3B, NDRG1 knockout also diminished significantly FGF-2 medicated ERK and AKT activation. The authors need to show the quantification in the original figure.
- d) It is Fig 4 kind of confusing, as there were no phosphor-VEGFR2 tyrosine specific antibody experiments. Authors need to show the various phosphor-Y specific antibody of VEGFR2 to come to any conclusion as such.
- e) Fig 5b, which phosphor-VEGFR2 antibody was used? It is not clear why siRNA of NDRG1 block both FGF-2 and VEGF-A mediated Ca-signaling however the further downstream signaling was VEGF-A selective---as per the authors conclusion.
- f) Fig 6a is very confusing and problematic. The IB control of NDRG1 is exactly same as the IP with PLC γ in response of VEGF-A lanes. Suggesting they might be in the same immunocomplexes without VEGF-A.
- g) Fig 6d is not convincing. It is hard to get any conclusion without quantitation.
- h) Fig 7 is an old data. There are several papers already published regarding using the same small molecule inhibitor of PLC γ to block angiogenesis. So, this data can be in supplement.

Responses to the Reviewers' Comments

Reviewer #1

Comment 1:

There is no data available regarding the purity of the mouse lung endothelial cells as well as the cell passages used for the experiments.

The authors indicated that endothelial cells were sorted with a CD31 antibody. However, in order to get pure endothelial cell populations it is required to conduct a second sorting, with an ICAM-2 antibody at the time of the first trypsinization when cells reach confluence.

Response:

(1) According to this comment, we performed the flow cytometric analysis using ICAM-2 antibody in addition to CD31 antibody to verify the purity of mECs isolated from *Ndrgr1*^{+/+} and *Ndrgr1*^{-/-} murine lung tissues (Supplementary Fig. 1). Independently established #1 and #2 mECs for each group certainly expressed both CD31 and ICAM-2 (Supplementary Fig. 1). Further, we also performed the western blotting analysis to check the expression of VE-cadherin (vascular endothelial cell marker) and Type I collagen (fibroblast marker) (Response to Reviewer only Fig. 1). WT-mECs strongly expressed VE-cadherin comparable to HUVECs, and there was no expression of Type I collagen in WT-mECs as well as HUVECs, suggesting that there was few contamination of fibroblasts in mECs. Together, these data strongly indicated that mECs established by our systems were purified vascular endothelial cells.

(2) We used mECs in all experiments ≤ 3 passages. Both of CD31 and ICAM-2 expressed in mECs from passage 1 to passage 3 (Supplementary Fig. 1).

We added above sentences in our revised manuscript. Thank you very much for your invaluable comment.

Comment 2:

In figure 5, it is not clear that NDGR-1 is required for PLC gamma phosphorylation. Quantitative data should be provided.

Response:

According to this comment, we examined the effect of NDRG1 knockout or silencing in ECs on PLC γ 1 phosphorylation in response to VEGF-A by western blot analysis for three times, and presented the graphs quantifying these results in Figure 5b and 5c of the revised manuscript. We confirmed that NDRG1 knockout or silencing in ECs decreased PLC γ 1 phosphorylation in response to VEGF-A (Fig. 5b and 5c). Thank you for your comment.

Comment 3:

In figure 6, the data showing recruitment of NDGR-1 and PLC-gamma to the same complex is not convincing. Additional experiments should be conducted and quantitative data should be provided.

Response:

This is a very important comment for the role of complex of NDRG1/PLC γ 1 in VEGF-A signaling pathway. In Fig. 6a and 6b, we showed that NDRG1 formed a complex with PLC γ 1 through its phosphorylation sites by IP assay and Pulldown assay. In this revised manuscript, we further examined whether interaction of NDRG1 with PLC γ 1 was important for VEGF-A-mediated phosphorylation of PLC γ 1 in ECs, and presented following relevant data.

(1) Transfection of GST- Δ 4 NDRG1 mutant, which could not interact with PLC γ 1 due to deleting phosphorylation sites, significantly decreased VEGF-A-induced PLC γ 1 phosphorylation compared with Mock and GST-NDRG1 transfection (Fig. 6c).

(2) In addition, proximity ligation assay (PLA), which is an antibody-based method to allow in situ detection of endogenous protein interaction and localization with high specificity and sensitivity, further confirmed the interaction of NDRG1 with PLC γ 1, and translocation of its complex to cell membrane in response to VEGF-A (Fig. 6e).

(3) Further, according to this comment, we quantitated the PLA spot on cell membrane when absence or presence of VEGF-A. Quantitative analysis of Fig. 6d and 6e of this revised manuscript showed that NDRG1/PLC γ 1 complexes were significantly recruited by VEGF-A on cell membrane in which activated PLC γ 1 induced hydrolysis of phosphatidylinositol 4,5-bisphosphate (PIP $_2$) to generate two second messengers, inositol 1,4,5-triphosphate (IP $_3$) and diacylglycerol (DAG) (Fig. 6d and 6e).

These data together strongly suggested that interaction of NDRG1 with PLC γ 1 accelerated PLC γ 1 phosphorylation in response to VEGF-A. Thank you for your comment.

Comment 4:

The authors do not provide any data on how the NDRG-1 antibody used for human samples was validated.

Response:

Prior to this experiments, we confirmed the specificity of NDRG1 antibody (ab124689, abcam) for human samples to stain NDRG1-positive and -negative human tumor samples which were previously classified as NDRG1 negative or NDRG1 positive by staining with anti-NDRG1 antibody (1:200, produced in our laboratory as cited in this manuscript #41) (Response to Reviewer only Fig. 2) (Kawahara A, et al., *Exp Ther Med.* 2011). Thank you for your comment.

Comment 5:

In page 15, lines 231-240, the data should be described more clearly.

Response:

According to this comment, we described more clearly these sentences in Results of this revised manuscript. Please confirm these sentences in page 14-15, lines 226-253. Please also refer to the response to comment 3. Thank you very much for your comment.

Reviewer #2

Comment 1 and 2:

One major problem of this manuscript is lack of developing the logistics of the experimental presentations and the Figures. For examples in Figure 1, the authors showed that RENCA cells which is highly expressing VEGF-A when implanted in mouse dorsal ear sac assays. But they didn't try to grow the cells for tumor growth, which is highly VEGF induced angiogenesis dependent. There animal model is the best model to test the hypothesis.

Similarly, Fig 2 started by showing that in breast cancer patients tissues expresses NDRG1 (by CD34

co-localization) in EC and then they started aortic ring experiments to show that role of NDRG1 in VEGF induced cell proliferation and migration in EC. Very hard to understand the logistic.

Response:

(1) Thank you very much for your invaluable comment. According to this comment, we totally reconstructed this revised manuscript, especially parts of Figs. 1 and 2. Please confirm our revised manuscript.

(2) To examine whether *NdrG1* knockout affects tumor growth and angiogenesis by RENCA cells, we have tried our best to perform the implantation experiments with RENCA in response to this comment during these six months. We implanted RENCA cells into the subcutaneous tissue of the right abdominal wall of the *NdrG1*^{+/+} and *NdrG1*^{-/-} mice by various methods such as modified cell concentration with or without 50% Matrigel. However, RENCA cells failed to form tumors with sufficient sizes in both *NdrG1*^{+/+} and *NdrG1*^{-/-} mice under all implanted conditions. This impaired tumor growth by RENCA may be due to difference of mice strain between our knockout mice on C57BL6 background and RENCA cells on BALB/c background. At present, we believed that two independent angiogenesis assays, dorsal air sac assay (Fig. 1c) and corneal micropocket assay (Fig. 1d, e), will consistently indicate that *NdrG1* deficiency impair VEGF-A-induced angiogenesis *in vivo*.

Comment 3:

Does VEGF treatment increase NDRG1 protein level within 7 min treatment? It is not clear why that was happened, as per Fig 3a. Which one is the real band of NDRG1? It is surprising that the authors didn't try to show the phosphorylation of tyrosine specific VEGFR2 in those experiments. In Fig 3B, NDRG1 knockout also diminished significantly FGF-2 medicated ERK and AKT activation. The authors need to show the quantification in the original figure.

Response:

(1) This comment is very important for the understanding of NDRG1 involvement in VEGF-A signaling pathway. We further examined whether treatment of VEGF-A increased NDRG1 protein levels for several times. Our further data with quantitative analysis consistently showed no significant enhancement of NDRG1 protein levels by VEGF-A (See also Fig. 3a of this revised manuscript).

Thank you very much for your comment.

(2) We and other groups previously reported that NDRG1 protein expression by immunoblotting showed the existence of multiple isoforms in various cells including cancer cells, with two major bands at 41 and 46kDa (Sahni S, et al., *Biochim Biophys Acta Mol Basis Dis.* 2019). Further, to elucidate which band was the specific band of NDRG1 detected by our western blotting analysis, we have performed NDRG1 specific silencing in HUVECs or human glioblastoma cell line U87MG using *NDRG1* siRNA (HSS173605, Invitrogen). Expressions of both bands were markedly downregulated by siRNA, consistent with our recent study on the role of NDRG1 in glioblastoma cells (Ito H, Watari K, Ono M, et al., *Cancer Research*, in press). Based on these data, we concluded that the both bands were NDRG1.

(3) According to this comment, we presented phosphorylation status of VEGFR2 (Y1175) in Fig. 3a of this revised manuscript.

(4) We compared the effect of NDRG1 knockout in ECs on expression of p-ERK1/2 and p-AKT by densitometry evaluation when treated with FGF-2, but there was no significant reducing effect on these proteins expression (please refer to Fig. 3b). According to this comment, we presented the quantitative data of expression of p-ERK1/2 and p-AKT in Fig. 3a and b of this revised manuscript.

Comment 4:

It is Fig 4 kind of confusing, as there were no phosphor-VEGFR2 tyrosine specific antibody experiments. Authors need to show the various phosphor-Y specific antibody of VEGFR2 to come to any conclusion as such.

Response:

According to this comment, we examined the expression of several VEGFR2 phosphorylation (Y951, Y996, Y1059 and Y1175) of ECs in Fig. 4d–f using anti-Y951 antibody (4491, Cell Signaling Technology), anti-Y996 (2474, Cell Signaling Technology), Y1059 antibody (3817, Cell Signaling Technology), and two Y1175 antibody (2478 and 3370, Cell Signaling Technology). VEGFR2 phosphorylation was maximized at 5 min to 10 min after VEGF-A treatment (Fig.3c, 3d, 5b, 5c, 5d).

Therefore, there was no apparent VEGFR2 phosphorylation at 40 min after VEGF-A treatment in the VEGFR2 internalization assay (Fig. 4d, e), and at 60 min after VEGF-A treatment in the VEGFR2 recycling assay (Fig. 4f). We presented brief procedure of these experiments as follow for your information;

VEGFR2 internalization assay (Fig. 4d and e): we incubated the cells with 0.2 mg/ml EZ-Link Sulfo-NHS-Biotin in PBS on ice for 30 min, quenched with cold PBS + 50 mM glycine. Then, we stimulated the cells with VEGF-A at 37 °C. At predetermined time points, we rinsed cells with cold PBS and incubated them for 15 min on ice with 225 mM Sodium 2-Mercaptoethanesulfonate to cleaved biotin moieties remaining at the cell surface. After 15 min, we incubated the cells for 10 min on ice with 900 mM iodoacetamide added.

VEGFR2 recycling assay (Fig. 4f): we stimulated HUVECs with VEGF-A for 30 min in the presence of cycloheximide (10 µM). For the assessment of VEGFR2 recycling, we additionally washed the cells 3 times with warm PBS and incubated them for 1 hr in 0.2% serum medium in the presence of cycloheximide (10 µM).

Comment 5:

Fig 5b, which phosphor-VEGFR2 antibody was used? It is not clear why siRNA of NDRG1 block both FGF-2 and VEGF-A mediated Ca-signaling however the further downstream signaling was VEGF-A selective---as per the authors conclusion.

Response:

(1) We used anti-phospho-VEGFR2 (Y1175) (2478, Cell Signaling Technology) in Fig.5b as described in Methods section. To make clear, we replaced “p-VEGFR2” with “p-VEGFR2 (Y1175)” in all Figures of this revised manuscript.

(2) In Fig. 5e, we showed that VEGF-A markedly increased cytoplasmic Ca²⁺ levels in siControl-treated hECs (orange line) up to 7 min, but had a very much smaller effect in si*NDRG1*-treated hECs (gray line) (Fig. 5e). In contrast, FGF-2 did not affect cytoplasmic Ca²⁺ levels in both siControl- (blue line) and si*NDRG1*-treated (yellow line) hECs (Fig. 5e), consistent with

previous published paper by other research group (McLaughlin AP & De Vries G, *Am. J. Physiol. Cell Physiol*, 2001, as cited in this manuscript #32). To make clear, we revised these sentences in Results of this revised manuscript.

Comment 6:

Fig 6a is very confusing and problematic. The IB control of NDRG1 is exactly same as the IP with PLC γ in response of VEGF-A lanes. Suggesting they might be in the same immunocomplexes without VEGF-A.

Response:

Thank you for your invaluable comment. All our data from Co-IP assay (Fig. 6a), Pulldown assay (Fig. 6b), subcellular protein fractionation assay (Fig. 6d), and PLA assay (Fig. 6e) indicated that NDRG1 already formed complex with PLC γ 1 in ECs without VEGF-A stimulation. Further, there was no significant increase of NDRG1/PLC γ 1 complexes when stimulated with VEGF-A. Thus, we presume that NDRG1 forms complex with PLC γ 1 before VEGF-A stimulation, and these pre-existence complexes mainly accelerated VEGF-A signaling pathway through PLC γ 1 phosphorylation and recruitment on cell membrane (Fig. 7).

Comment 7:

Fig 6d is not convincing. It is hard to get any conclusion without quantitation.

Response:

According to this comment, we newly presented the quantitative analysis of PLA spot around cell membrane with or without VEGF-A in Fig. 6e of this revised manuscript. Quantitative analysis showed that PLA positive spots were significantly augmented around cell membrane in response to VEGF-A (Fig. 6e). Thank you for your comment.

Comment 8:

Fig 7 is an old data. There are several papers already published regarding using the same small

molecule inhibitor of PLCg to block angiogenesis. So, this data can be in supplement.

Response:

According to this comment, we transfer these data from Fig.7a–c to Supplementary Fig. 3 of this revised manuscript. Thank you for your comment.

Reviewers' comments:

Reviewer #1 (Remarks to the Author):

Point 1: the authors did not provide data regarding the purity of the endothelial cells isolated from mouse lungs. They conducted FACS analysis for endothelial markers only. Which percentage of cells used in the experiments are endothelial cells?

We ask you to please answer this question in the text.

Point 3: No quantification for the co-IP studies has been provided (Figure 6a). The recruitment of NDGR-1 and PLC-gamma to the same complex (co-IP) is not convincing.

We ask that you quantify the co-IP data and note in the text the limitations of the experiment and the findings if any.

Quantification for figure 5d, 5g and 5h is lacking. The conclusions taken from these experiments are unclear in the text. We ask you to please quantify and discuss the results.

Responses to the Reviewers' Comments

Reviewer #1

Point 1: the authors did not provide data regarding the purity of the endothelial cells isolated from mouse lungs. They conducted FACS analysis for endothelial markers only. Which percentage of cells used in the experiments are endothelial cells?

We ask you to please answer this question in the text.

Response:

According to this comment, we added percentage of CD31 or ICAM-2 positive cells in Results of this revised manuscript (Page 9). Thank you very much for your comment.

Point 3:

No quantification for the co-IP studies has been provided (Figure 6a). The recruitment of NDGR-1 and PLC-gamma to the same complex (co-IP) is not convincing.

We ask that you quantify the co-IP data and note in the text the limitations of the experiment and the findings if any.

Quantification for figure 5d, 5g and 5h is lacking. The conclusions taken from these experiments are unclear in the text. We ask you to please quantify and discuss the results.

Response:

According to this comment, we quantified Figure 5d, 5g, 5h, and 6a, and presented the quantitative data of these experiments in this revised manuscript. Further, we also discussed these data in Results and Discussion of this revised manuscript. Please confirm these sentences. Thank you very much for your invaluable comment.